# Early detection of critical urban events using mobile phone network data

**Pierre Lemaire**[1], **Angelo Furno**[1]*, **Stefania Rubrichi**[3], **Alexis Bondu**[3], **Zbigniew Smoreda**[3], **Cezary Ziemlicki**[3], **Nour-Eddin El Faouzi**[1], **Eric Gaume**[2]

**1** LICIT-ECO7 UMR T9401, ENTPE, University Gustave Eiffel, Lyon, France, **2** GERS, University Gustave Eiffel, Nantes, France, **3** Orange Innovation, Châtillon, France

* angelo.furno@univ-eiffel.fr

**Data Availability Statement:** Network signaling data are proprietary and confidential. We obtained access to these data from Orange France within the framework of the research project ANR DISCRET (ANR-19-FLJO-0002-01). For the sake of

## Abstract

Network Signalling Data (NSD) have the potential to provide continuous spatio-temporal information about the presence, mobility, and usage patterns of cell phone services by individuals. Such information is invaluable for monitoring large urban areas and supporting the implementation of decision-making services. When analyzed in real time, NSD can enable the early detection of critical urban events, including fires, large accidents, stampedes, terrorist attacks, and sports and leisure gatherings, especially if these events significantly impact mobile phone network activity in the affected areas. This paper presents empirical evidence that advanced NSD can detect anomalies in mobile traffic service consumption, attributable to critical urban events, with fine spatial (a spatial resolution of a few decameters) and temporal (minutes) resolutions. We introduce two methodologies for real-time anomaly detection from multivariate time series extracted from large-scale NSD, utilizing a range of algorithms adapted from the state-of-the-art in unsupervised machine learning techniques for anomaly detection. Our research includes a comprehensive quantitative evaluation of these algorithms on a large-scale dataset of NSD service consumption for the Paris region. The evaluation uses an original dataset of documented critical or unusual urban events. This dataset has been built as a ground truth basis for assessing the algorithms' performance. The obtained results demonstrate that our framework can detect unusual events almost instantaneously and locate the affected areas with high precision, largely outperforming random classifiers. This efficiency and effectiveness underline the potential of NSD-based anomaly detection in significantly enhancing emergency response strategies and urban planning. By offering a proactive approach to managing urban safety and resilience, our findings highlight the transformative potential of leveraging NSD for anomaly detection in urban environments.

## Introduction

As recent studies on Mobile Crowd Sensing (MCS) suggest, the detection of urban anomalies and the subsequent adaptation of event management strategies in case of emergency are becoming more realistic [1, 2]. MCS is part of the opportunistic detection paradigm,

reproducibility of the research, a minimal subset is available at https://github.com/licit-lab/discret/tree/main/data-sample, as agreed in the project Data Management Plan. Access to the full dataset can be requested from Orange on a contractual basis, by contacting the Director of the Research Augmented Customers and Collaborators Domain at Orange Innovation/Research (currently: Thierry Nagellen, email: thierry.nagellen@orange.com). Please note that Thierry Nagellen is not affiliated with this research as an author.

**Funding:** This work is supported by the French ANR research projects DISCRET (grant number ANR-19-FLJO-0002-01) and PROMENADE (grant number ANR-18-CE22-0008). EG is the author awarded for the project DISCRET. AF is the author awarded for the project PROMENADE. URL of the funder: https://anr.fr The funders had no role in study design, data collection and analysis, decision to publish, or preparation of the manuscript.

**Competing interests:** The authors have declared that no competing interests exist.

leveraging the massive amount of data passively generated by users from their use of mobile devices [3]. In particular, the ability to study changes in the *urban metabolism* via MCS has emerged with the generalized use of mobile phones.

Many services and activities in our daily lives are mediated through the use of mobile phones. The data resulting from their usage open new opportunities to sense individual behaviors, improving the insight into human mobility [4] or communications [5] for instance. These insights can be turned into actionable information to address other important questions in fields such as urban and transportation planning [6–9] and population census [10].

Beyond the investigation of the regular daily activities of individuals, MCS has also proved to be a valuable resource for the analysis of anomalous or critical situations such as life-threatening epidemic outbreaks [11–13], major urban infrastructure failures, natural or societal disasters (e.g., earthquakes [14–17], floods [18] and conflicts [19]). Such critical events have a deep impact on human dynamic models and procedures that are designed essentially on and for stationary situations [20–23]. These unusual conditions induce stress and can significantly disrupt the normal operation of urban or national infrastructures, potentially precipitating their failure [24]. Urban crowds represent both fragile and exposed entities, and a high potential for major disruption and cascading effects that are difficult to manage. Meanwhile, they are a potential and efficient way of detecting anomalies.

In such contexts, mobile phones effectively function as *in-situ* sensors, capable of capturing real-time alterations in individuals' mobility and communication patterns in response to emergencies or atypical conditions. Understanding how individuals' behavior changes when exposed to rapidly evolving or unknown conditions is challenging [25, 26]. However, real-time analysis of the volume of data flows on mobile phone networks could help to identify and localize early incidents or emergency situations, and/or augment the data from other more conventional sources (emergency calls, video surveillance, etc.).

In their seminal work, Bagrow et al. [25] have shown that voluminous Call Detail Records (CDR) from Mobile Network Operator (MNO) embed valuable indications on how people react to major emergencies (terrorist attacks, plane crashes, earthquakes and major failures of urban infrastructures) as well as non-catastrophic major urban events (music festivals, sporting events). More specifically, they have shown that high-risk situations provoke a huge increase in communication activity (phone calls and text messages) when compared to the everyday activity, within a limited area around the event. The volume of calls begins to decrease immediately after the emergency, indicating that the tendency to communicate is strongest at the beginning of the event. On the other hand, festive events, which attract large crowds, show a more gradual increase in the communication activity, quite different from that of *jump-decay* type observed in threatening emergency situations. On the spatial side, the amplitude of the anomaly is stronger in the vicinity of the event and decreases rapidly, with an exponential decrease relative to the distance from the epicenter and the nature of the event. In particular, life-threatening events affect the observed call volume up to tens of kilometers, while other anomalies are limited to the urban or peri-urban scale.

Building on the conclusions of Bagrow et al. [25], researchers have explored the potential of CDRs to characterize the consequences of tragic events, mostly on a very large scale such as a country. For instance, Lu et al. [14] have analyzed the displacement of 2 million cell phone users in Haiti, to assess the predictability of their migration to safer areas in case of earthquakes and to quantify the expected population decrease in the affected areas for the months following a disaster. The temporal analysis of several mobility and presence indicators calculated from the mobile phone data (e.g., estimated radius of gyration, entropy of frequently visited places) confirmed that a better understanding of the behavior of people affected by disasters is possible with mobile phone data. Similar conclusions have been reached for earthquakes [17], floods

[27] and large scale events, such as concerts, festivals and demonstrations [28]. This improved understanding can potentially help decision makers to simulate and forecast the number of evacuated people within urban areas in limited time and cost.

On the other hand, the literature on using such data for the timely detection of events is limited. More specifically, to the best of our knowledge, solutions have not been proposed for application in operational contexts related to urban monitoring, particularly for supporting the early detection of critical urban events over very short temporal scales (i.e., minutes) and with precise location (i.e., tens to hundreds of meters). For instance, Dobra et al. [29] are among the first authors to propose a CDRs-based methodological framework for the automatic detection of emergency situations in Rwanda. Leveraging an extensive database of emergency and non-emergency events, they qualitatively test the performance of the proposed detection system at a daily temporal resolution, which successfully captures anomalous behavioral patterns associated with a wide range of events. Even though their work allows identifying days with anomalous calling and mobility behavior, the solution does not possess the necessary time granularity and reactivity to support the rapid and online detection of critical events. From a spatial perspective, the authors consider an extremely coarse granularity, corresponding to a grid cell unit of 25 km$^2$. More generally, while existing approaches based on CDRs can provide valuable insights, they often exhibit limited spatial [30] and temporal resolution [31] and are typically conducted retrospectively rather than in real-time within urban contexts. CDRs are, in fact, processed information that is only available with a certain delay.

As an additional limitation, existing solutions rely solely on qualitative and often subjective evaluations of the methods' capacity to pinpoint anomalous behavior, missing a quantitative and objective characterization of the capability of such methods to accurately detect critical events. This reliance on qualitative assessment means that the effectiveness and reliability of these methods cannot be rigorously measured or compared, leading to potential biases and inconsistencies in the evaluation process. In conclusion, these limitations preclude meaningful comparisons with existing solutions for several reasons:

1. **Temporal Resolution**: Most existing methods operate on a daily temporal resolution, which is insufficient for the timely detection of events that unfold over minutes.

2. **Spatial Resolution**: The granularity of spatial data in current approaches is often too coarse to precisely localize critical urban events.

3. **Qualitative Evaluation**: The absence of quantitative metrics means that existing methods are not rigorously validated, making it difficult to benchmark their performance against new approaches.

4. **Operational Context**: Many proposed solutions have not been tested in real-world urban dense contexts, which limits their practical applicability and reliability.

By addressing these issues, our proposed method aims to provide a robust, real-time detection framework with fine-grained spatial and temporal resolution, ensuring both the accuracy and practical applicability of the solution.

Specifically, to overcome these limitations, it is proposed herein to base the early detection of urban anomalies on the analysis of a richer form of mobile phone data, namely the Network Signalling Data (NSD). NSDs contain detailed information about calls, text messages, data session initiations and terminations, and network control events, collated on the fly and used by the cell phone operators for the supervision and management of their cell phone networks. NSDs could theoretically be made available and processed in real time, offering an extremely

fine spatial and temporal granularity. The dataset analyzed hereafter is anonymized and contains only the numbers of service requests (calls, text messages, data sessions, etc.) aggregated on a per-minute and per-antenna basis. This unique high-resolution data enables precise tracking and analysis of mobile network activity. Moreover, the NSD used in this study are related to a dense urban territory, the city of Paris, thus providing a rich and diverse dataset that is representative of a complex urban environment. By focusing on Paris, our study leverages the complexities of a major metropolitan area to validate the effectiveness of our detection framework in dense urban settings. The assumption on which the detection method is based is that anomalous cell phone service request numbers are either related to rare infrastructure malfunctions or to real-life unusual or critical events such as transportation accidents, spontaneous demonstrations, natural hazards, etc. The early detection and location of the latter could accelerate the deployment of emergency units in case of serious, life-threatening incidents.

The main contributions of this paper are the following:

- The potential of early anomaly detection based on NSDs is explored.

- Two existing anomaly detection methods have been enhanced and tested using a large real-world NSD dataset provided by a cell phone operator, as well as an ad hoc developed comprehensive quantitative evaluation framework.

- A database of documented urban events has been created to assess the relevance and effectiveness of the proposed detection methods.

## Materials and methods

### Data source

The Mobile phone data used in this study were provided by Orange France, the main mobile network operator in France, in the form of time series of aggregated NSD. The time series were extracted based on the processing of signaling messages exchanged between mobile devices and the mobile network, usually collected by network probes to allow monitoring and optimizing the mobile network activities. Such messages are triggered by a variety of network events, such as (i) voice and texting communications, (ii) handovers (i.e., transferring of ongoing call or data session from a cell to a new one), (iii) Location Area (LA) and Tracking Area (TA) updates (i.e., cell changes that cross boundaries among larger regions named LA in 2G/3G and TA in 4G), (iv) active paging (i.e., periodic requests to update the location of the device started from the network side), (v) network attaches and detaches (i.e., devices joining or leaving the network as they are turned on/off), and (vi) data connections (i.e., requests to assign resources for traffic generated by mobile applications running on the device). They enable a time-stamped localization, at cell level, of any user of the Orange mobile network, with a resulting level of geographic accuracy that depends on the density of cells (i.e. antenna) coverage in the considered area.

Data collected for the purposes of this study concern about 27 million daily Orange subscribers in the metropolitan area of France (12 TB of data on average per day) and cover a three-month period, from March 15 to June 15 2019. They were aggregated by event type (hereafter called services), at cell level over a one-minute resolution. More precisely, given a cell or a group of cells of the network $\mathcal{A}$ of size $m$, a set of mobile network services $\mathcal{S}$ of size $k$ (typically, calls in 3G or 4G, text messages, data transfers, hand-overs, etc.), we aggregate their activity (count of requests) per minute over a given period of time $\mathcal{T}$ of $n$ minutes. The aggregation guarantees the anonymity of network users since only the total volume of activity is analyzed, and the labels of the individual phones are not considered.

The resulting dataset can be modelled as follows:

$$s_a(t) = (v_a^0(t), v_a^1(t), \ldots, v_a^{k-1}(t)) \tag{1}$$

where:

- $a \in \mathcal{A}$ is the id of a cell

- $t$ is the minute (timestamp) of an observation in $\mathcal{T}$

- $v_a^s$ is the number of events for a given service $s \in \mathcal{S}$ for the cell $a$

- $k = |S|$ is the number of considered mobile network services.

Intuitively, we propose a detection approach based on the hypothesis that unusual events such as accidents or crowd gatherings generate a volume of mobile phone activities significantly distinct (i.e. higher) from the everyday regular activity. For instance, witnesses of an accident or participants to a demonstration, may call for rescuers, film incidents and post images on social networks, and share their status and position with relatives through text-messages, thus generating a temporary increase of the mobile phone activity.

## Network anomalies and urban events

Interpreting network anomalies in the context of concurrent real-world critical events is a complex and multifaceted challenge. Network anomalies can arise from a broad spectrum of urban circumstances, each carrying distinct implications for network operation.

Events like accidents or attacks tend to be highly localized, generating significant cellular activity, such as phone calls, immediately after their onset. We refer to these events as *punctual*, marked by a specific location and a clear start time, though their conclusion may not be as easily determined, depending on factors such as the event's severity and response efficiency.

Crowd gatherings, particularly those not occurring on a regular basis, are also likely to prompt unusual mobile phone activity. This activity can range from highly localized (e.g., concerts or sports games) to widely spread (e.g., climate hazards or spontaneous celebrations), with the temporal dynamics varying significantly.

Moreover, while events like concerts, festivals, and special sports occasions typically generate increased network activity without posing direct threats to participants, their classification as anomalies can be ambiguous. Automated detection systems may variably classify such events as anomalies, but both outcomes can be deemed acceptable. Nonetheless, distinguishing potential life-threatening situations within these events is crucial, necessitating anomaly characterization not only by their state but also by intensity. Given the absence of comprehensive data on a broad array of real-world scenarios, a deeper analysis of these events exceeds the scope of this study.

Our primary focus hereafter is not on the interpretation of real-world events through network anomalies as observed via NSDs but on illustrating the effectiveness of our proposed automatic detection systems in identifying anomalies that frequently correlate with a broad spectrum of significant real-world occurrences.

The taxonomy presented in Table 1 offers a contextual framework that highlights the variety of events affecting network activity, thereby showcasing the efficacy of our anomaly detection approaches. By concentrating on the automatic identification of anomalies within this taxonomy, we aim to demonstrate the ability of our methodologies to recognize a diverse array of events, from unforeseen disasters to scheduled gatherings, emphasizing the importance of network data analysis in urban dynamics comprehension and response. This direction encourages future research to delve into the complex relationships between detected network

**Table 1. A taxonomy of anomalous events depending on their spatial and temporal spread.**

| Temporal spread | Spatial spread | |
|---|---|---|
| | **Local** | **Widespread** |
| sudden onset, fuzzy end | explosion, fire, accident, terrorist attack... | earthquake, spontaneous gathering, riot... |
| fuzzy start, fuzzy end | concert, sport game, exhibition... | climatic hazard, demonstration, running, cycle race |

anomalies and their actual world counterparts, enhancing the precision and utility of automatic anomaly detection for practical applications.

## The reference anomaly database

To conduct a quantitative assessment of the proposed methodologies, we compiled a Database comprising 350 past Uncommon Events (DBUE) pertaining to the entire city of Paris. This database represents a benchmark used for the computation of performance metrics, such as precision and recall. The establishment of this database marks, per se, a contribution of our work, filling a notable void in ground truth data available for validation throughout the reference period. In the process of assembling this database, we encountered unavoidable subjective challenges, including the assessment of an anomaly's relevance, scale, and duration. To counteract these challenges and reduce subjectivity, we conducted an exhaustive review of news reports on demonstrations and significant incidents. Furthermore, we employed a strategy of independent database construction by multiple contributors, ensuring a more robust and varied analytical perspective.

Based on the taxonomy introduced in Table 1, each Uncommon Event (UE) of our anomaly database is defined by the following features:

- **Geographic Coordinates** representing the epicenters of the event. While most events are accurately depicted with a single pair of coordinates (e.g., sports games, concerts), others like demonstrations or races may span multiple locations. For such events, a concise list of key geographic points is used, including start and end points, and locations of notable incidents along the event's path.

- **Temporal Information**. This includes the starting date and time of the event. Where applicable, an ending date and time are also provided, which is common for events with a defined duration like concerts or sports games. For events lasting several days (e.g., festivals), we specify an interval of days along with the start and end times that apply uniformly across all days.

- **Descriptive Information**. A textual summary of the event, accompanied by links to news reports and the source, if available, to provide context and additional details.

Overall, our DBUE covers 86 events occurred between March 18th, 2019 and June 15th, 2019, in Paris. It includes the following Uncommon Events (UEs): 38 concerts, 24 sports events (football games, running races, etc.), 20 cultural events (ceremonies, fairs), 1 demonstration, 2 domestic fires and the infamous Notre-Dame fire. Only 2 events (the May 1st demonstration and the Marathon of Paris) were spread across multiple locations within our reference area.

As a result of the previous definitions, the DBUE is not directly comparable with the Detected Aanomalies (DAs) produced by either of the detection methodologies. This is due to the following main reasons:

- UEs are defined *event-wise* (which includes an interval of time) while DAs are defined minute-wise, which is the smallest temporal granularity available in our MND dataset.

- DAs only happen on antenna locations, while UEs are unlikely to occur at such accurate locations.

- Multiple DAs on close antenna locations can correspond to a single UE.

Therefore, when assessing the detection methods, DAs are matched to UEs based on specific spatial and temporal tolerance levels. The tolerance levels adopted to identify a match between a UE and a DA are defined as follows:

- On the spatial scale, for each geographic epicenter of an UE, we consider a radius centered on its coordinates (typically from a few meters to a few hundred meters, depending on the density of the mobile network). This spatial tolerance is used to identify the set of closest antenna locations associated to DAs falling in the allowed range.

- On the temporal scale, we use the time interval already present in the definition of the UE, when available. When the UE is defined only in terms of a starting time, the temporal interval is defined by the maximum acceptable time for a detector to identify the unusual behavior. In our experiments, we assume this tolerance limit to be at most 15 minutes after the (real) beginning of the event. An additional tolerance interval may be set before and after the event, to account for detection that may be linked to crowd gathering prior and posterior to the event (typically, attendees leaving a concert or a game).

## Anomaly detection: General features

The idea behind an online anomaly detector in a temporal series is firstly to forecast the nominal activity $\tilde{v}_a^s(t)$. The observed deviations $\epsilon_a^s(t)$ from this nominal behavior can then be used as a measure of the magnitude of the anomaly. Approaches can differ in both the nominal behavior forecasting model and the detection method of the anomalous observed deviations from the nominal behavior.

The deviation in a time series at a given instant *t* from the nominal activity may be expressed as follows:

$$\epsilon_a^s(t) = v_a^s(t) - \tilde{v}_a^s(t) \tag{2}$$

where

- $v_a^s(t)$ is the observed activity level with $a \in \mathcal{A}$, $s \in \mathcal{S}$ and $t \in \mathcal{T}$,

- $\tilde{v}_a^s(t)$ is the corresponding predicted nominal activity level,

- $\epsilon_a^s(t)$ is the deviation between the observed and the predicted activities, usually referred to as *innovation* in signal processing.

The detection may be based directly on the value of $\epsilon_a^s(t)$ or on the estimated probability of exceedance of the value $\epsilon_a^s(t)$. The two approaches considered in this paper implement univariate prediction models and deviation computation. It means that the methods compute nominal values for each service $s \in \mathcal{S}$ independently. Following the approach from our previous work [32], a simple statistical fusion scheme is used when anomaly detection is based on the activity of several cell phone services. Table 2 summarizes the characteristics of the two approaches featured in this paper. Further details are provided in the next two sections.

**Table 2. Summary of the methods compared in this study.**

| Method | Activity prediction | Deviation and Detection |
|---|---|---|
| Signatures | Median of weekly activities (same weekday, same time) filtered temporally through a Butterworth filter. | Deviation is a signed difference to the prediction. A Gamma survival function is fitted on the distribution tail for each service. The final estimated deviation, a probability of exceedance, is a product of all survival functions. |
| Adaptive | For a given service, a forecasting model based on past observed values [33] is calibrated. | The forecasting error is monitored using an adaptive control chart. |

## The signature method

The first method considered herein is an enhanced version of the method proposed by some of the authors in [32]. It is based on the definition, for each cell and service of the phone network $(a, s)$, of a median, nominal, weekly fluctuation of activity: a signature $r_a^s$. The signature is defined based on a series of training weeks $W_{train}$, if possible not including unusual events.

Considering the service volume data per minute $v_a^s(w, m_T^w)$ where $w \in W_{train}$ in the training dataset and $0 \le m_{T^w} < 10080$ a minute of week $w$, the corresponding weekly signature is computed as $r_a^s(m_T^w) = \mathcal{B}(\hat{\mu}(\{v_a^s(w, m_T^w) | w \in W_{train}\}))$, where $\hat{\mu}(\cdot)$ is the median operator and $\mathcal{B}$ a Butterworth filter, applied to the ordered series of minute-wise median values. The signature $r_a^s$ being defined, the set of observed typical deviations from the signature $\epsilon_a^s(w, m_T^w) = \{v_a^s(w, m_T^w) - r_a^s(m_T^w) | w \in W_{train}\}$ can be computed for the training data set, for a each cell, service $(a, s)$ and time step $m_T^w$. It is hypothesized herein that the statistical distribution of the deviations is identical for all minutes in the week.

A single statistical distribution can then be adjusted to the whole set of observed deviations $\epsilon_a^s(w, m_T^w)$. If necessary, this hypothesis could be relaxed and the statistical model complexified in the future. The calibrated cumulative statistical distribution of the deviations is a compound distribution combining (a) the empirical distribution of frequent observed deviations and (b) a Gamma distribution, specifically tailored to fit the subset of the rarest deviations. To delineate these rare deviations, we employ the formula $\epsilon_a^s = \mu_a^s + h \cdot \sigma_a^s$, where $\mu_a^s$ denotes the mean and $\sigma_a^s$ the standard deviation of the entire sample of deviations $\epsilon_a^s$. Here, $h$ is a predetermined constant set to 2.32, chosen with the intent to isolate deviations that fall in the extreme upper percentile of the distribution—theoretically, the top 1% under a Gaussian distribution assumption.

This method of selection is premised on identifying deviations that are significantly larger than the norm, indicative of potentially anomalous behavior warranting closer examination. However, the application of this threshold in practice captures approximately 3% of the training deviation samples, rather than the expected 1%. This discrepancy reveals that the actual distribution of deviations from the network behavior possesses heavier tails than those of a Gaussian distribution, implying a higher prevalence of extreme deviations. Such an observation underscores the necessity of employing a Gamma distribution to accurately model these rare but critical deviations, as it is more adept at accommodating the observed distribution's tail behavior.

By distinguishing between common and rare deviations in this manner, the calibrated distribution function effectively segments the data into those deviations that are typical and expected versus those that are atypical and potentially indicative of significant network anomalies.

In the implementing stage (i.e., testing phase), the likelihood (probability of exceedance) of any observed deviation from the signature $\epsilon_a^s(w, m_T^w)$, $w \in W_{test}$ can be estimated according to

the calibrated cumulative distribution function of the deviations: $\mathcal{L}_{\epsilon_a^s(w,m_T^w)} = P[X \geq \epsilon_a^s(w, m_T^w)]$. The detection of anomalies is based on fixed thresholds for this likelihood. In cases where the detection is based on several services, it is proposed hereafter to simply compute a compound likelihood as the product of the individual likelihoods for each service $\prod_a^{s \in \mathcal{S}} \mathcal{L}_{\epsilon_a^s(w,m_T^w)}$ for cell $a$ and minute $(w, m_T^w)$. It is worth noting that the product operation theoretically implies the independence of the deviations observed for the various services, a condition that is in fact not met. The product of likelihood can hardly be interpreted as a probability, and the anomaly detection thresholds are therefore not theoretically derived, but adapted for each tested combination, to maintain similar proportions of anomaly detection in each tested situation. This limitation of the current approach may be further investigated in the future. Of course, the whole procedure can only be implemented for mobile phone services and network cells with a significant amount of activity, requiring filtering of those where such conditions are not met.

## The adaptive method

The second tested approach combines a time-series forecasting model based on the Facebook Open Source Prophet library [34] and an *adaptive* Shewhart control chart [35] for the detection of anomalous deviations between predicted and observed cell-phone service activities.

The selected forecasting model is an additive forecasting model, inspired by Generalized Additive Models (GAM) [36]. It decomposes the time-dependent signal into non-periodic and periodic components of three types—trend (non-periodic changes), seasonality (a technical term that refers to any periodic change), and holidays (abruptly irregular patterns occurring over one or more days). Markov Chain Monte Carlo methods are implemented to calibrate the model [33]: i.e., one model is calibrated for each cell and service or combination of services. The selected model accounts for a weekly-periodic signal and uses school holidays as additional (additive) regressor.

The series of resulting deviations between predicted $\tilde{v}_a^s(t)$ and observed $v_a^s(t)$ activity levels is monitored using an adaptive Shewhart control chart. Control charts are one of the most frequently used procedures in statistical process control to detect anomalies. They have been widely used in quality engineering [37] and then extended to areas including healthcare [38–40], economics, informatics [41], and environment [42, 43]. A Shewhart control chart is a simple method based on the estimated mean $\mu_a^s$ and standard deviation $\sigma_a^s$ of past realizations of the monitored process (i.e., the prediction deviations $\epsilon_a^s$ in this study). At each time step, the computed deviation is considered anomalous if $\epsilon_a^s \geq \mu_a^s + h \cdot \sigma_a^s$, and consequently the chart is not updated. The parameter $h$ has been set equal to 3 hereafter.

The originality of the implemented approach is the update of $\mu_a^s(t)$ and $\sigma_a^s(t)^2$ over time, their computation being based on exponentially decaying weighting function $\omega$ of past observed deviations $\epsilon_a^s$ with decaying rate $\lambda$:

$$\mu_a^s(t) = \frac{\sum_{n=0}^{\infty} \omega^{t-n} \epsilon_a^s(t-n)}{\sum_{n=0}^{\infty} \omega^{t-n}} \tag{3}$$

$$\sigma_a^s(t)^2 = \sum_{n=0}^{\infty} \omega^{t-n} \left(\epsilon_a^s(t-n)\right)^2 - \left(\sum_{n=0}^{\infty} \omega^{t-n} \epsilon_a^s(t-n)\right)^2 \tag{4}$$

being $\omega = 2^{-\lambda(t-n)}$, $\lambda = \log_2\left(\frac{1}{2}\right)/\tau$, and $\tau$, also known as *half-life*, a user parameter which determines how quickly the past is forgotten. For the tests presented hereafter, $\tau$ was set to 1440 minutes or 24 hours. This update trick allows for a more efficient implementation since only

three scalar values need to be stored in RAM at each step: $\sum_{n=0}^{\infty} \omega^{t-n} \epsilon_a^s(t-n)$, $\sum_{n=0}^{\infty} \omega^{t-n}$, and $\sum_{n=0}^{\infty} \omega^{t-n}(\epsilon_a^s(t-n))^2$.

Moreover, the adaptive nature allows the anomaly detection method to adapt to unusual situations that lead to a temporary concentration of population in given areas due to special events such as festivals, fairs, and sports events. The corresponding local drastic increase in cell phone activity may induce the detection of anomalies over the whole duration of the event and mask critical events that may occur if the detection method is not tuned accordingly.

## Anomaly levels and thresholds definition

Both methods provide an anomaly measure or likelihood, which depends on the distribution of the errors on each antenna, as well as the fusion method, described in the section dedicated to the signatures method when several services are analyzed simultaneously. The difference in methods, the absence or presence of certain services for certain antennas as well as the fusion process, not taking account of an eventual inter-service correlation, make it difficult to compare anomaly levels based on the sole anomaly likelihood value.

For this reason, we introduced anomaly levels and their corresponding thresholds to reflect the scarcity of an event magnitude. Levels are defined in terms of anomaly frequency, similarly to the method used to measure the magnitude of floods: such as a 100-year flood, i.e., an event that has a 1% probability of being equaled or exceeded in any given year [44, 45].

More specifically, based on the training dataset, we defined three error or likelihood thresholds, corresponding to magnitudes that are exceeded, on average, once every 4 hours, 1 day and 1 week, respectively. In other terms, the level 1 threshold is the separation corresponding to the $(4 \times 60)^{th}$ (respectively $(24 \times 60)^{th}$, $(7 \times 24 \times 60)^{th}$) quantile. This is computed directly on the distribution of the training data, for each antenna $a$, after the fusion step when several services are used.

As the outputs of the proposed methods, we decided to generate alerts indicating an anomaly level at every minute for each antenna within the designated territory, referred to as DAs. These DAs, are treated as independent entities across both time (*minute-wise*) and space (*antenna location-wise*).

Furthermore, we categorize the anomaly level using a discrete scale with four possible values. A level of 0 indicates that the antenna $a \in \mathcal{A}$, at minute $t \in \mathcal{T}$, exhibits typical behavior and is not considered anomalous. Conversely, an anomaly level of 1 (respectively 2, 3) signifies a minor (respectively medium, major) deviation from the norm. Based on the previous thresholds' definition, on average, anomalies of level 1 (respectively levels 2, 3) are expected to occur approximately once every 4 hours (respectively once per day, once per week) in nominal situations, providing a framework for understanding the relative frequency and severity of these events.

## Illustration with some case studies

To evaluate the effectiveness of the two approaches under consideration, they were tested using real-world data provided by Orange France (the leading French mobile phone company) for the period from March 15 to June 15, 2019. We selected a set of antennas corresponding to the whole Paris city, and implemented a train/test strategy to obtain detection for the whole dataset. We partitioned the 3 months dataset into 3 distinct roughly 1-month test sets, two sets being selected in turn for training of the methods and the remaining for the testing. This allowed us to have independent training and testing sets, while performing testing the detection efficiency over the whole dataset.

Our dataset encompasses several real-world anomalous events of significance, enabling us to assess the relevance of our approach from a qualitative perspective. Below, we highlight two such events, employing the signature-based approach to illustrate the detection process. Comparable outcomes were observed using the adaptive method, underscoring the robustness of our findings.

The first one is the fire that ruined the roof of the Notre-Dame cathedral in Paris as well as its spire, on April 15th in 2019 (see Fig 1). The fire alarm started at 18:20, interrupting a mass. A wrong location was initially investigated by guards, which led firemen being summoned late at 18:51. They arrived within 10 minutes. Smoke started becoming visible from the outside of the cathedral at 18:52. From there, our detection system (Fig 1a reporting the anomaly likelihood levels over time) shows a rapid surge in calls and SMS, which ends up producing a persistent, high level alarm within 5 minutes after the first visible signs of fire at the closest antenna to the cathedral location (Fig 1c). The alarms then spread quickly to neighboring antennas for several hours, which illustrates the spatial accuracy of the detection approach (Fig 1d).

The second situation is a demonstration that took place in Paris on May 1st (Fig 2).

The international Workers' Day in France in 2019 was highly symbolic. It followed a series of violent encounters between police and protestersthe previous year. It was also amidst the Yellow Vests movement crisis, which was also frequently subject to documented violent riots for several months already. Thus, the Unions' May 1st 2019 demonstration was highly followed, and subject to a vast live coverage from various media, as well as massive anti-riot

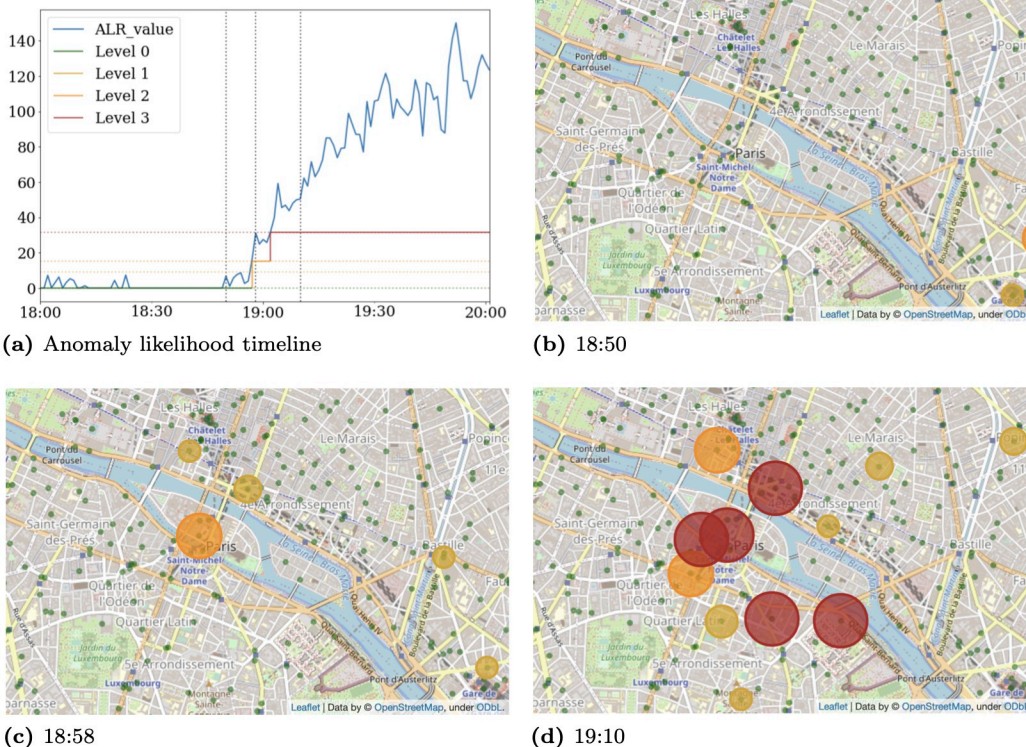

**(a)** Anomaly likelihood timeline                    **(b)** 18:50

**(c)** 18:58                                          **(d)** 19:10

**Fig 1. Timeline for the Notre-Dame fire on April 15th 2019. (a)** shows the anomaly level observed on the closest outdoor antenna to the cathedral. **(b)**, **(c)** and **(d)** show the corresponding map at different times, centered at the cathedral location. Disk sizes and colors are linked to the alarm level. Green spots correspond to antenna locations. Maps throughout this research article were created using open-source data and tiles from OpenStreetMap and the OpenStreetMap Foundation, which is made available under the Open Data Commons Open Database License (ODbL).

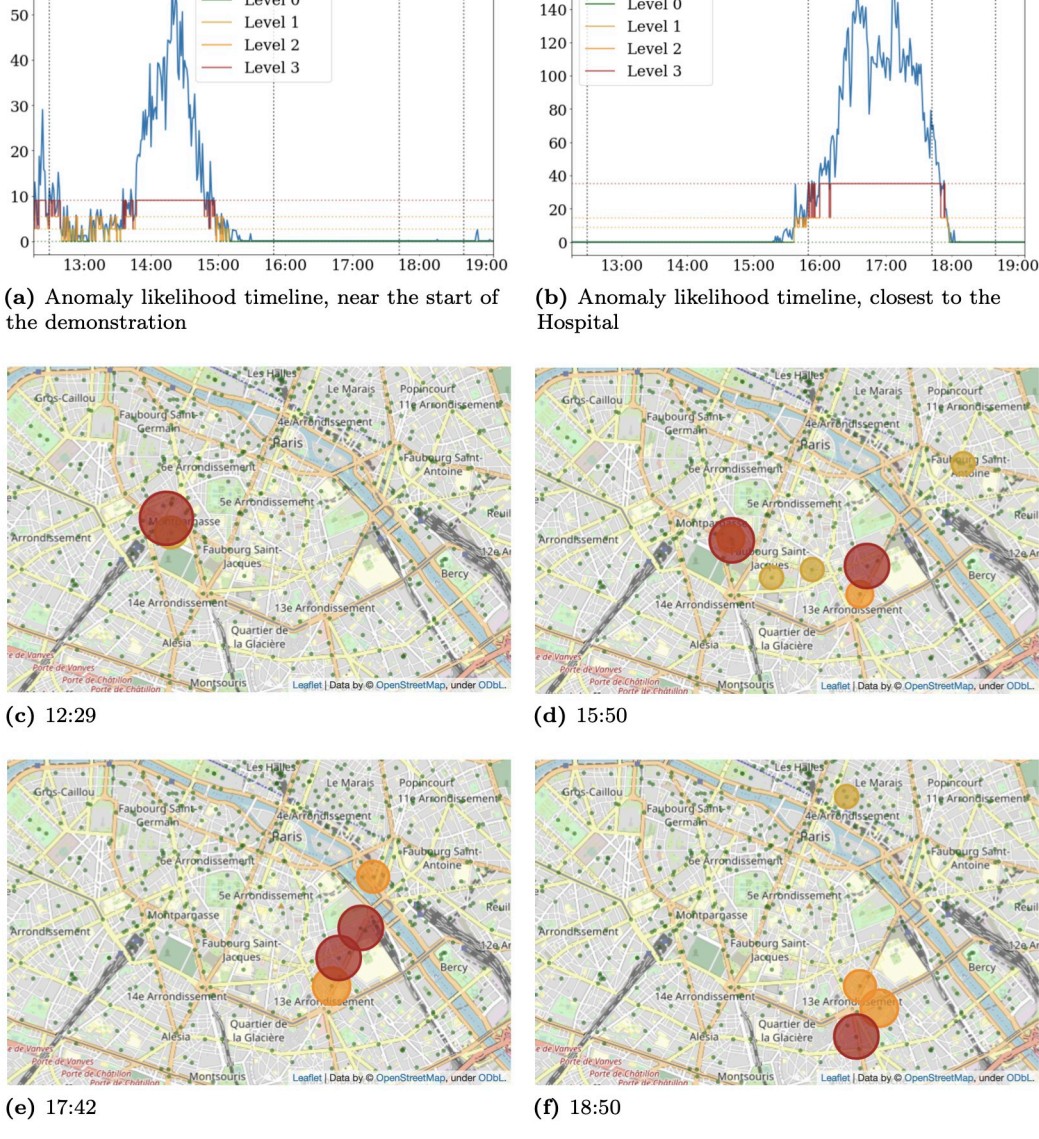

**Fig 2. Timeline for May First demonstration.** a and b show the anomaly levels observed on key locations (start of the demonstration, also visible on c) and the hospital (highest alarm level common to d and e). c, d, e and f show the corresponding map at different times. Disk sizes and colors are linked to the alarm level. Green spots correspond to antenna locations.

police presence. Many protesters with various motivations were present, including Yellow Vest protesters as well as Black Blocks. Violent encounters happened along the demonstration way, including the so-called invasion of the hospital La Pitié Salpétrière by protesters.

Interestingly, as reported in Fig 2, the sequence of antennas detected as anomalous by our solution over time corresponds to the path taken by the protesters during the demonstration. High anomaly levels are concentrated close to the locations where the most significant incidents were reported. More notably, some specific events were detected by the system before being reported in the media's live coverage of the day. The northernmost anomaly shown at 17:42 (e), which is persistent, occurred very close to a fire reported at 18:21 by the live coverage

of "Le Monde" a prominent national journal. This delay is likely explained by the journalists' need to source, verify, prioritize, and contextualize their information before releasing.

## Performance indicators

The aforementioned spatio-temporal criteria allow building the confusion matrix reported in Table 3. This table can be used to match each DA produced by any of the two detection methods to the set of UEs from our ground-truth anomaly database. For instance, a DA is considered as a *True Positive* (TP), when its duration is *included* inside the time range defined above the beginning of the UE *and* its spatial location is *inside* the spatial tolerance associated to the location of the UE.

The confusion matrix is computed over the reference testing periods, to calculate the Precision and Recall indicators, which characterize the response of a detector.

Precision (the proportion of true positive results in all positive predictions) is given by the equation:

$$\text{Precision} = \frac{TP}{TP + FP} \tag{5}$$

Recall (the proportion of true positive results in all actual positives) is given by the equation:

$$\text{Recall} = \frac{TP}{TP + FN} \tag{6}$$

Precision is calculated for each minute of the test period, allowing every minute detected as anomalous to be categorized as either a true or a false positive based on the spatio-temporal matching rules. This definition of precision is thus relatively straightforward to compute and relevant from a semantic perspective, as the level of precision correctly decreases with increasing amounts of false positives over the reference period. However, the application of recall at a minute-level granularity encounters two main issues. First, long-lasting events covering large areas disproportionately influence the overall recall despite not necessarily being the most significant. Second, the presence of multiple alarms within the duration of an event is not essential for the detection system to be deemed effective. Ideally, with high precision, even a single alarm per event could suffice for the detection to be considered successful, highlighting that recall may be more meaningfully assessed on an event basis rather than indiscriminately by minute.

To address these issues, we adopted an *Event-based* Recall, in addition to the Minute-wise definition. An event is marked as successfully detected if at least *n* alarms of level *l* are issued within the spatio-temporal boundaries of the event. In our evaluation, we set *n* = 1, implying that a single alarm of any specified level is considered sufficient for the successful detection of an event. This assumption holds under conditions of high precision and a generally low alarm frequency.

Conversely, adapting the precision metric to event granularity proves challenging. In the event-based granularity, a "successful detection" must be redefined to consider the detection

**Table 3. Confusion matrix.**

| | UE interval | |
| --- | --- | --- |
| | **Inside** | **Outside** |
| DA | True Positive (TP) | False Positive (FP) |
| No DA | False Negative (FN) | True Negative (TN) |

of entire events, not just individual alarms. This implies the aggregation of possible multiple alarms into discrete events. Effectively clustering alarms or signals into broader categories of detection involves the development of a new detection system or an additional processing layer to the methodologies explored in this study. While this represents a significant area of future interest, it falls outside the current paper's scope.

We considered three criteria for the quantitative evaluation of the proposed event detection methods: 1) the minute-wise precision, 2) the minute-wise recall, and 3) the event-wise recall. We evaluated the two proposed detection methodologies (Sec. Anomaly detection: general features) for various alarm levels, examining how changes in sensitivity affect the trade-off between precision and recall. A highly sensitive detector tends to increase recall at the expense of precision and vice versa. Capturing this balance is crucial for characterizing the detector's performance for specific applications. Due to the extensive size of the dataset and the non-homogeneous definitions of the precision/recall indicators, a detailed sensitivity analysis was challenging.

Particularly, our analysis focused on three distinct sensitivity levels corresponding to the three thresholds introduced in Sec. Anomaly levels and thresholds definition. We decided not to use the classical F-Score as an aggregate performance metric, as precision and recall were computed at different granularity.

Finally, we contextualized precision and recall against a hypothetical detector that issues detection randomly with a frequency $r$ corresponding to the selected thresholds. This classifier, denoted as *No-Skill detector* in the following, serves as a performance baseline to assess the significance of the alarms generated by our system.

Specifically, the precision of the No-Skill or random detector is simply equal to the ratio between the cumulative duration of the anomalous events for each cell of the network in our UE database ($|\mathcal{GT}^+|$) and the total observed time period ($\mathcal{T}$):

$$\text{Precision}_{\text{No-Skill}} = \frac{|\mathcal{GT}^+|}{\mathcal{T}}. \tag{7}$$

The No-Skill minute-wise Recall is obviously equal to the detection frequency $r$, as minutes corresponding to events will be sampled alike all other minutes of the observed time period at this frequency $r$ by the random detector:

$$\text{Recall}_{\text{No-Skill}} = r. \tag{8}$$

## Results

This section details the results of the quantitative evaluation of the two methodologies outlined in Sec. Anomaly detection: general features, aiming to compare their effectiveness and understand the influence of different parameters and configurations. To facilitate the reproducibility of our research findings, all code used in the analysis, alongside a representative sample of the anonymized network signaling data for a subset of antennas in the vicinity of Notre-Dame cathedral over the three-month period, is available in the publicly accessible GitHub repository: https://github.com/licit-lab/discret/tree/main/data-sample.

### Evaluation of the tested methods

Table 4 reports the performance results of both the Signature and the Adaptive detection methods. The No-Skill Precision for our UE database corresponds to 0.23%, which is 5 times less than our worst recorded result (1.03% for the adaptive method when using the level 1

**Table 4. Performance indicators for the studied methods.** $R_{min}$ (resp. $R_{ev}$) stands for Recall at minute (resp. event) granularity.

| Method | Level | $R_{min}$ (%) | Recall$_{No-Skill}$(%) | Precision (%) | $R_{ev}$ (%) |
|---|---|---|---|---|---|
| Signature (4 metrics) | ≥1 | 4.24 | 0.15 | 2.38 | 80.23 |
| | ≥2 | 3.57 | 0.029 | 9.27 | 44.19 |
| | ≥3 | 3.20 | 0.006 | 28.63 | 36.05 |
| Adaptive (4 metrics) | ≥1 | 6.05 | 0.15 | 1.03 | 66.28 |
| | ≥2 | 4.84 | 0.029 | 3.16 | 54.65 |
| | ≥3 | 2.21 | 0.006 | 7.02 | 26.74 |

error threshold). The best Precision score reached is 28.63%, which is more than 100 times better than a No-Skill detector. This shows that despite all mentioned limitations, the detection methods are effective and that our database is of some significance even if far from perfect. Another indication of the relevance of both our methods and our protocol is that the Recall (in Minute and Event granularity) and Precision are linked to the sensitivity as expected: the higher the sensitivity, the higher the Recall but the lower the Precision. Lastly, we notice a difference in behavior between both methods. At lower detection levels, the Adaptive method yields significantly better Recall performances than the Signature method, especially at minute granularity. However, the behavior is different at higher detection levels. The Precision in particular is very different and advantageous for the Signature method at higher levels.

To elucidate and contrast the behaviors of the proposed detection methods and to determine the optimal sensitivity setting, traditional approaches would recommend plotting a ROC-Curve. However, in situations of significant class imbalance, as it is the case with our dataset, ROC curves may not effectively represent the data characteristics, potentially exaggerating the imbalance effect. Consequently, we have opted to analyze Precision-Recall curves, following recommendations from literature that suggests PR curves offer a more accurate depiction in imbalanced contexts [46].

The effectiveness of a method is indicated by the area under the curve, with a larger area denoting higher efficiency. In generating such a curve, we introduced three additional intermediate anomaly detection thresholds, corresponding to detection frequencies of 8 hours, 12 hours, and 2 days to complement the 3 already existing thresholds and refine the sensitivity spectrum. This inclusion aims to provide a comprehensive view of how each method performs across a range of operational scenarios, from frequent to rare event detection thresholds.

Fig 3 delineates a clear superiority of the less complex Signature method at the event granularity, as it consistently outperforms the Adaptive method across all detection levels. At the minute granularity, results are varied; the Adaptive method shows superiority at lower detection levels in terms of recall, attributed to its enhanced stability in error measurement. Particularly, at the highest detection level, the Signature method excels in Precision, likely due to its ability to discount unstable antennas (those with extremely high anomaly rates) with null signatures, a mechanism not implemented in the Adaptive approach.

## Ablation study: Contribution of various elements and services

Our initial hypothesis posited that emergency situations would significantly increase mobile phone call volumes. Contrary to this, findings from the qualitative study detailed in Sec. Illustration with some case studies indicated the importance of utilizing a combination of services. This section delves into the individual and collective contributions of various services to validate our assumption and assess the efficacy of a fusion method.

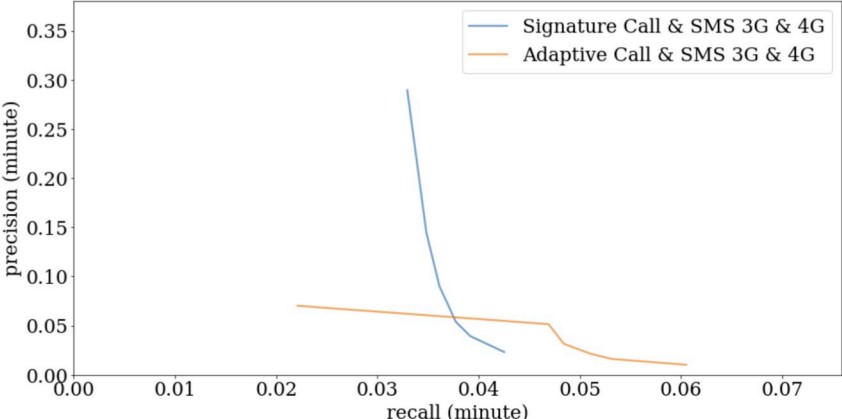

**(a)** PR curve at minute granularity

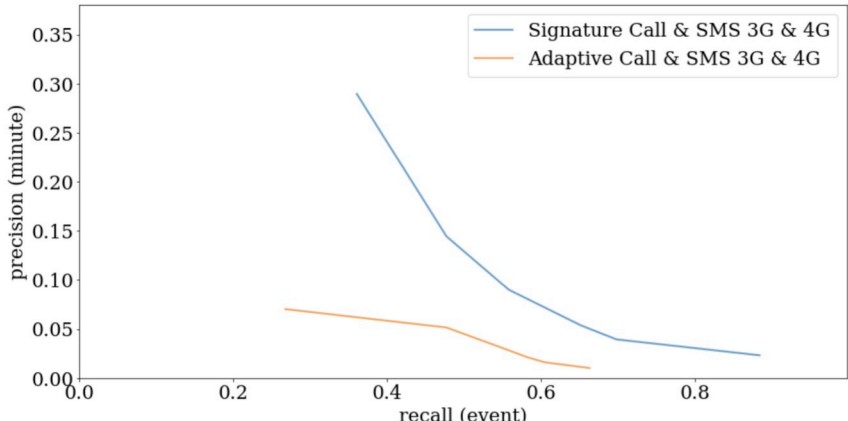

**(b)** PR curve at event granularity

**Fig 3. Precision Recall curves at minute and event granularity, for the Signature and the Adaptive methods fusing 4 services: Call and SMS, both in 3G and 4G.**

We concentrate our analysis on the Signature method, which exhibited superior performance and stability in our evaluations. The study encompasses four distinct services from our dataset: 3G Calls, 4G Calls, 3G SMS, and 4G SMS, alongside their integration, including the fusion of 3G and 4G Calls, 3G and 4G SMS, and all four metrics combined (as previously illustrated in Fig 4).

Rather surprisingly, the analysis revealed that individual 3G services, particularly SMS, marginally outperformed random chance, despite being utilized four times less frequently than their 4G counterparts. These services predominantly contributed noise. Interestingly, merging 3G services with their 4G equivalents did not detrimentally affect outcomes, suggesting the robustness of our fusion method. This indicates that 3G services, despite their noise, can complement the information provided by 4G services.

Differences in performance were noted between granularity. Notably, 4G SMS consistently held significant informative value, performing exceptionally well at event granularity and enhancing results when combined with 4G Calls. This suggests that in densely populated areas, SMS becomes a preferred mode of communication. Given the dataset is from 2019, it is

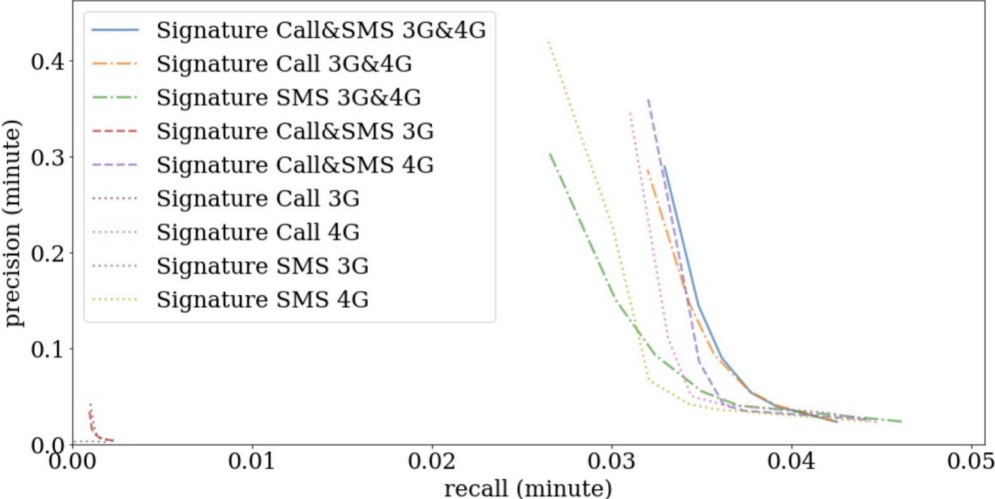

**(a)** PR curve at minute granularity

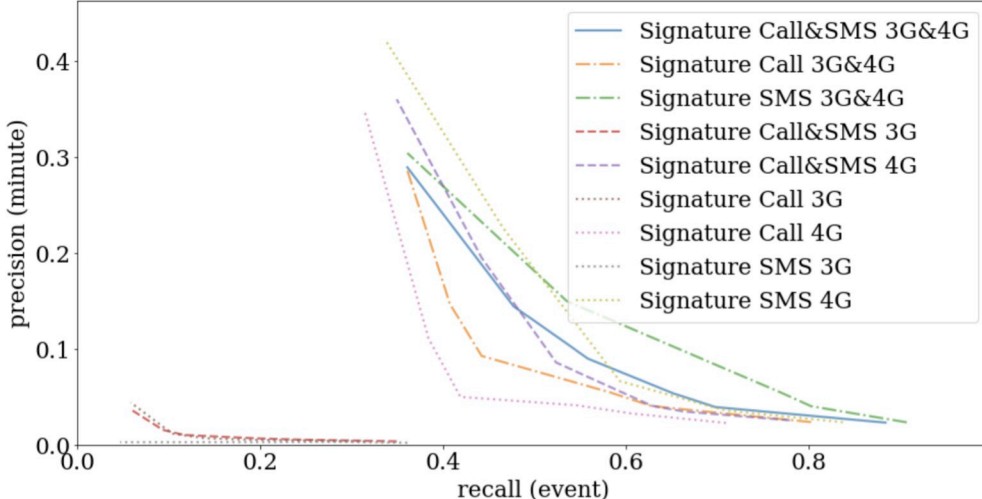

**(b)** PR curve at event granularity

**Fig 4. Precision Recall curves at minute and event granularity, for the Signature method only, with individual outputs for Call and SMS services in 3G and 4G (dashed curves) and the fusion between several configurations.**

plausible that modern encrypted messaging services might now serve as complements or alternatives to traditional SMS.

At minute granularity, Calls displayed greater accuracy, reinforcing the notion that integrating multiple services yields improved outcomes. The results at event granularity, however, were more nuanced, suggesting a stronger association between events and alarms when analyzing 4G SMS volumes.

## Discussion

This exploratory work sheds light on the complexities of detecting unusual events through mobile service data (network probe signaling data more precisely), revealing both the potential and the limitations of the approach.

Our contribution could have numerous relevant applications to support the services provided by the following city stakeholders:

1. **Emergency Response Forces**: The potential of NSD for rapid and precise identification of critical events can significantly improve the response time of emergency services by providing real-time alerts about potential critical events. This timely information allows for the rapid deployment of resources to affected areas, potentially saving lives and mitigating damage.

2. **Crisis Management Authorities**: The integration of NSD and social media data can help crisis managers gain a more detailed understanding of unfolding events. This comprehensive situational awareness is crucial for making informed decisions and coordinating efforts among different emergency response units.

3. **Urban Security Agencies**: For agencies responsible for maintaining public safety, the ability to detect anomalies at a finer spatial and temporal resolution enables more precise monitoring of high-risk areas. This capability can prove vital during large-scale events like the Olympic Games, where the potential for security threats is elevated.

4. **Local Government and Municipal Services**: Local authorities can use the data to monitor urban dynamics continuously, allowing for proactive measures in traffic management, public transportation adjustments, and crowd control during critical events.

5. **Public Health Officials**: In the event of public health emergencies, such as disease outbreaks or environmental hazards, the system can provide early warnings that facilitate rapid public health responses. The integration of various data sources ensures a comprehensive understanding of the situation, aiding in the containment and management of such crises.

However, it must be emphasized that the proposed approach is not intended to be autonomous. It is conceived as a complementary early warning system, augmenting other monitoring tools based on emergency call centers, surveillance cameras, and corroborative on-site reports. Consequently, city stakeholders can leverage the large-scale, precise, and timely detection capabilities provided by NSD to enhance overall preparedness and response efficiency.

To be effective as a pre-alarm, the method must be very reactive (detection typically within a few minutes after the onset of the event) and must provide an accurate localization. It should not be over-sensitive and provide, if possible, the intensity of the detected anomalies, which may reflect the magnitude of the event and help the users prioritizing their response.

Both tested methods for anomaly detection, and particularly the signature-based one, have proven adept at leveraging mobile service data to identify UEs, showcasing the benefits of fusing data across various services. Although alarms based on mobile calls correlate strongly with our UE database, SMS-based alarms provide a wider event coverage, albeit with reduced spatial and temporal accuracy. This suggests that integrating findings across different data granularity offers a more comprehensive understanding of network behavior during UEs. However, our current fusion approach makes the simplifying assumption of independence among the different mobile services, treating these signals as independent to simplify the calculation of the compound likelihood of observed deviations from the signatures. This assumption can be a limitation, as signaling generated by different mobile operator services can be correlated and differently influenced by the nature of the underlying event. This independence assumption may limit the accuracy of our anomaly detection methodology, as it does not account for the potential dependencies between services.

Another consideration, deriving from our quantitative evaluation, relates to instances where our methods failed to detect certain local events, such as fires or car accidents with limited number of witnesses. This limitation underscores the importance of further research, particularly into the dynamics of signal variation during such incidents. Predominantly, our current methods excel at identifying events characterized by large gatherings, pointing to a need for a broader analytical scope.

Additional limitations derive from the ground truth dataset of anomalies used in the quantitative analysis and from the approach adopted to match DAs and UAs, as further detailed below.

The simplicity of our anomaly annotation approach facilitates the definition of a ground truth reference dataset for quantitative evaluation and comparison purposes but struggles with disambiguating specific scenarios. Notably, it fails to capture crowd motion or intensity variations within anomalous events due to the independent treatment of spatial and temporal coordinates. This limitation should be considered when utilizing the database for training or evaluative purposes.

Moreover, the collation of unusual events based on various sources is an almost endless task and is certainly far from exhaustive. The resulting database can hardly encompass all potential real-life incidents that could atypically affect the mobile network, such as minor car accidents or private events. Thus, the UE database serves best as a comparative tool for evaluating different detection systems, their real skills being difficult to evaluate.

The interpretation of Precision and Recall metrics requires careful consideration due to several factors:

- The overlap between UEs and DAs is inherently imprecise, complicating the spatial and temporal definition of UEs. A lack of alerts during a UE might indicate its insufficiency to impact mobile network activity significantly.

- Treating all event minutes equally may not accurately reflect the fluctuating intensity of UEs.

- The UE database's exclusion of several real-life events that could justify alerts underscores fundamental documentation gaps.

- The significant class imbalance necessitates a cautious comparison of Precision due to the typically low values of such a metric. This could suggest random alert generation by the detector or limited applicability of the UE database.

## Conclusion and future research directions

In this study, we address the challenge of detecting unusual events in urban environments by leveraging aggregated network signaling data provided by the French leader mobile phone operator on a per-minute and per-antenna basis. Our innovative approach capitalizes on the spatial and temporal richness of such dataset, preserving user anonymity while offering insights into both routine and emergency scenarios. Despite the inherent complexity of processing large-scale urban data, and building upon our previous work [32], we have proposed two alternative approaches that can be deployed in real-time for effective anomaly detection.

Our findings demonstrate the reactivity of these strategies in identifying significant unusual events, usually within minutes from the start of the anomalous event. Furthermore, we propose a quantitative, city-wide evaluation framework based on an extensive Unusual Events database, enabling the benchmarking of detection methods and hyperparameter optimization. The evaluation underscores the utility of aggregated network signaling data in identifying

crowd-related anomalies with notable temporal and spatial precision and recall, despite substantial class imbalances. Additionally, our analysis reveals the complementary nature of information from diverse services, such as short messaging and voice calls, in enhancing anomaly detection.

Our study reveals several limitations that provide avenues for future research. Firstly, future work should focus on improving the sensitivity and specificity of detection algorithms to handle a wider range of event magnitudes. Current methods, both static (Signature approach) and dynamic (Adaptive approach), may overlook subtle yet informative anomalies beneath certain thresholds. Enhancing the synergy between different telecommunications services, particularly the different roles of 3G, 4G, and 5G communication, as well as the contextual usage of calls, SMS, and Internet service data patterns, presents another avenue for exploration. For instance, it seems promising to overcome the simplifying assumption of independence among the different mobile services, and to explore more sophisticated statistical fusion models that account for the dependencies between services. For example, copula or multivariate distribution models, as well as deep learning solutions, fit on a large historical dataset could provide a more accurate representation of the joint distribution of deviations across services. Additionally, we are currently exploring the possibility of cross-correlating mobile phone data with other data sources, specifically Twitter data, in real-time. This integration is expected to provide more specific nuances on the nature of events, enhancing our ability to differentiate between various types of anomalies and improving the accuracy and reliability of our detection methodologies with information derived from additional data sources. Moreover, preliminary qualitative analysis suggests that visual tools can vividly highlight anomalies through the aggregation of single-antenna detections, hinting at the potential for developing a more robust detection system through clustering techniques. This direction, along with a deeper understanding of service interplay, will guide our efforts to refine anomaly detection in urban environments.

Secondly, our reliance on qualitative evaluations highlights the need for exploring additional metrics and extensive ground truth databases that can allow for more objectively assessing the performance of detection methods. Implementing robust validation frameworks will enhance the reliability and comparability of different approaches.

Thirdly, the dynamic nature of urban environments necessitates adaptive models capable of real-time learning and updating. Research should explore machine learning, particularly deep learning techniques, that can continuously adapt to new data patterns, thereby improving the robustness and accuracy of anomaly detection systems. For instance, LSTM-based models are particularly relevant for forecasting nominal values for each service, as they have proven to be efficient for time series simulations in numerous domains (e.g., [47]).

Lastly, integrating additional data sources such as social media feeds and IoT sensor networks could enrich the context of detected anomalies, providing a more holistic view of urban events. Exploring multi-modal data fusion techniques will be a critical step towards achieving comprehensive urban monitoring solutions.

## Author Contributions

**Conceptualization:** Angelo Furno, Zbigniew Smoreda, Nour-Eddin El Faouzi, Eric Gaume.

**Data curation:** Pierre Lemaire, Angelo Furno, Stefania Rubrichi, Zbigniew Smoreda, Cezary Ziemlicki.

**Formal analysis:** Pierre Lemaire, Angelo Furno, Stefania Rubrichi, Zbigniew Smoreda, Eric Gaume.

**Funding acquisition:** Pierre Lemaire, Angelo Furno, Nour-Eddin El Faouzi, Eric Gaume.

**Investigation:** Pierre Lemaire, Angelo Furno, Stefania Rubrichi, Zbigniew Smoreda, Eric Gaume.

**Methodology:** Pierre Lemaire, Angelo Furno, Stefania Rubrichi, Alexis Bondu, Nour-Eddin El Faouzi, Eric Gaume.

**Project administration:** Angelo Furno, Zbigniew Smoreda, Eric Gaume.

**Resources:** Angelo Furno, Cezary Ziemlicki, Eric Gaume.

**Software:** Pierre Lemaire, Angelo Furno, Stefania Rubrichi, Alexis Bondu, Cezary Ziemlicki.

**Supervision:** Angelo Furno, Zbigniew Smoreda, Eric Gaume.

**Validation:** Pierre Lemaire, Angelo Furno, Stefania Rubrichi, Alexis Bondu, Zbigniew Smoreda, Nour-Eddin El Faouzi, Eric Gaume.

**Visualization:** Pierre Lemaire, Angelo Furno, Eric Gaume.

**Writing – original draft:** Pierre Lemaire, Angelo Furno, Stefania Rubrichi, Eric Gaume.

**Writing – review & editing:** Pierre Lemaire, Angelo Furno, Stefania Rubrichi, Alexis Bondu, Zbigniew Smoreda, Nour-Eddin El Faouzi, Eric Gaume.

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
