## [Decision Letter · Decision Letter 0]

24 May 2024

PONE-D-24-11293Early Detection of Critical Urban Events using Mobile Phone Network DataPLOS ONE

Dear Dr. Furno,

Thank you for submitting your manuscript to PLOS ONE. After careful consideration, we feel that it has merit but does not fully meet PLOS ONE’s publication criteria as it currently stands. Therefore, we invite you to submit a revised version of the manuscript that addresses the points raised during the review process.

In particular, for the paper to be accepted, the authors need to address the following items:

a) Better situate the reader with respect to how the current paper compares to existing literature.

b) Provide some suggestions on how the results can be used by stakeholders involved in emergency response planning.

We look forward to receiving your revised manuscript.

Kind regards,

Luan Carlos de Sena Monteiro Ozelim, D.Sc.

Academic Editor

PLOS ONE

“This work is supported by the French ANR research projects DISCRET (grant number 627

ANR-19-FLJO-0002-01) and PROMENADE (grant number ANR-18-CE22-0008).”

“This work is supported by the French ANR research projects DISCRET (grant number

ANR-19-FLJO-0002-01) and PROMENADE (grant number ANR-18-CE22-0008).

EG is the author awarded for the project DISCRET. AF is the author awarded for the project PROMENADE.

URL of the funder: https://anr.fr

6. We note that Figures 1 and 2 in your submission contain [map/satellite] images which may be copyrighted. All PLOS content is published under the Creative Commons Attribution License (CC BY 4.0), which means that the manuscript, images, and Supporting Information files will be freely available online, and any third party is permitted to access, download, copy, distribute, and use these materials in any way, even commercially, with proper attribution. For these reasons, we cannot publish previously copyrighted maps or satellite images created using proprietary data, such as Google software (Google Maps, Street View, and Earth). For more information, see our copyright guidelines: http://journals.plos.org/plosone/s/licenses-and-copyright.

a. You may seek permission from the original copyright holder of Figures 1 and 2 to publish the content specifically under the CC BY 4.0 license. 

7. Please remove your figures from within your manuscript file, leaving only the individual TIFF/EPS image files, uploaded separately. These will be automatically included in the reviewers’ PDF.

Reviewers' comments:

Reviewer's Responses to Questions

**Comments to the Author**

1. Is the manuscript technically sound, and do the data support the conclusions?

Reviewer #1: Yes

Reviewer #2: Yes

2. Has the statistical analysis been performed appropriately and rigorously? 

Reviewer #1: N/A

Reviewer #2: Yes

3. Have the authors made all data underlying the findings in their manuscript fully available?

Reviewer #1: Yes

Reviewer #2: Yes

4. Is the manuscript presented in an intelligible fashion and written in standard English?

Reviewer #1: Yes

Reviewer #2: Yes

5. Review Comments to the Author

Reviewer #1: The paper focuses on utilizing network signaling data (NSD) from mobile phone networks to detect anomalies in mobile traffic service consumption, enabling the early detection of critical urban events. The study aims to provide actionable information for urban and transportation planning, population census, and emergency response by analyzing mobile phone data,. The paper discusses the importance of understanding how individuals' behavior changes in response to emergencies and unknown conditions, highlighting the role of mobile phones as in-situ sensors. Various methodologies for real-time anomaly detection in mobile traffic service consumption are introduced, with a focus on the effectiveness of different detection thresholds and fusion methods. Overall, the paper presents a strong technical foundation and provides valuable insights into the potential of using NSD for anomaly detection in urban environments. However, there are some minor concerns:

1. Providing a detailed comparison with existing methods would help situate the research within the broader context of the field.

2. Including a discussion on the limitations of the proposed methodologies and potential areas for future research could enhance the paper's completeness.

3. Consider expanding on the implications of the findings for urban planning and emergency response strategies to provide a more comprehensive understanding of the practical applications of the research.

Reviewer #2: The article aims at identifying the occurrence of relevant city events, positive like concerts or negative like fires, leveraging the analysis of anonymized sequences of mobile network signaling usage patterns.

The difficulties in correlating the statistical detection of anomalies in signaling trends with the correspondent specific event happening in the city is touched, and probably is partially due also to the constraints of the analyzed signaling information (e.g. data user communication patterns are currently not only related to the operator services, like voice calls or SMS, but today leverage also a plethora of Internet based services, that are less “visible” at Mobile Operator signaling level).

Signaling generated by the usage of different Mobile Operator’ services appear to be considered as substantially independent, whereas in some cases different services could be correlated (e.g. number of paging and number of calls) or can be differently influenced by the type of event (e.g people asking for help during a terroristic attack inside a building could prefer texting in respect to a more audible voice call).

Authors, anyway, do not target the ambitious recognition of a specific event, but mainly “array of events”, and presented results are honestly recognized as “far from perfect” by authors itself.

Minor editorial comments:

• Does “To”, in “To of data on average per day”, mean Terabyte?

• What is the meaning (pgrf "Conclusion") of the “?” in “…upon 604 our previous work [? ]...” ?

• The acronym “DA” seems to be defined in page 9, but used also before definition.

6. PLOS authors have the option to publish the peer review history of their article (what does this mean?). If published, this will include your full peer review and any attached files.

Reviewer #1: No

Reviewer #2: **Yes: **Davide Micheli

---

## [Author Response · Author response to Decision Letter 0]

9 Jul 2024

Response to the Reviews and Decision

Title:

Early Detection of Critical Urban Events using Mobile Phone Network Data

Manuscript Reference Number:

PONE-D-24-11293

Authors:

 • Pierre Lemaire

 • Angelo Furno* (corresponding author)

 • Stefania Rubrichi

 • Alexis Bondu

 • Zbigniew Smoreda

 • Cezary Zmielicki

 • Nour-Eddin El Faouzi

 • Eric Gaume

Date:

July 2, 2024

Message from the Authors

Dear Editors and Reviewers,

We greatly appreciate your constructive feedback, which has significantly enhanced the quality of our manuscript. We have thoroughly addressed your comments and incorporated your valuable suggestions into the revised version. Specifically, we have highlighted the key contributions of our work, better positioned our paper within the context of related literature, and clarified the potential practical applications and benefits for stakeholders. We have also incorporated all the editing and formatting changes requested to comply with the journal’s format and have generally improved the quality of our paper as suggested by the reviewers. To facilitate your review, the updated sections are marked in red in one of the revised versions of the manuscript.

Each comment has been addressed separately in the detailed response below. The comments we received are boxed, and our responses follow each comment. All page and reference numbers in our response are based on the revised manuscript unless otherwise stated. The page and reference numbers mentioned in the reviewers’ comments are based on the original manuscript. The references used to create our review responses are listed in the reference section on the last page of this response document.

We look forward to your feedback and hope that you find the revised manuscript satisfactory.

Sincerely,

the Authors.

Response To Editor

Overall Comments

After careful consideration, we feel that it has merit but does not fully meet PLOS ONE’s publication criteria as it currently stands. Therefore, we invite you to submit a revised version of the manuscript that addresses the points raised during the review process.

Response

We would like to thank the Editor for collecting and relaying the review responses to us. We are resubmitting the revised manuscript with the aim of more clearly highlighting the contributions of this work as well as addressing the points raised during the review process.

Editor’s Comment

In particular, for the paper to be accepted, the authors need to address the following items:

a) Better situate the reader with respect to how the current paper compares to existing literature.

Response

This work is situated in the literature on the use of mobile crowd sensing and, more specifically, mobile network operator data to study changes in urban metabolism when exposed to unfamiliar events. In the revised Introduction section, we have analyzed more in details the current research on the potential of Call Detail Records (CDRs) to capture human behavior changes in such anomalous contexts. Most of these studies have focused primarily on understanding and analyzing the predictability of human collective behavioral patterns after the critical event has occurred, with the aim of eventually improving the response. Only a limited number of studies have investigated the use of such data for the timely detection of events themselves, and, to the best of our knowledge, no study has considered the usage of Network Signalling Data (NSD) for this purpose.

The main novelties of our study with respect to the existing literature are thus:

 1. Focus on Timely Detection: While existing research predominantly addresses post-event analysis to understand human behavior, our study shifts the focus towards the timely detection of critical events. This proactive approach is crucial for improving rapid response and mitigation strategies in urban areas.

 2. Utilization of Network Signalling Data (NSD): Unlike traditional studies that rely on CDRs, our research leverages another form of mobile phone data, namely NSD, which provide a much finer spatial and temporal resolution. This granularity, as proven in our study, allows for real-time monitoring and more accurate detection of anomalies, which is not feasible with the coarser data granularity of CDRs.

 3. Development of a Quantitative Evaluation Framework: Existing solutions often rely on qualitative assessments of their effectiveness. In contrast, we have developed an ad hoc quantitative evaluation framework that objectively measures the performance of the proposed detection methods. This framework ensures rigorous validation and comparison against benchmarks, providing a more reliable assessment of its capabilities.

 4. Towards Operational Applicability in Dense Urban Areas: Our study demonstrates the relevance of NSD for rapid and precise detection of critical urban events and evaluates two alternative solutions for the practical applicability of our methods in real-world urban monitoring contexts. We address the limitations of previous studies that have not been comprehensively and quantitatively evaluated in terms of performance, especially in dense urban environments. By showcasing the feasibility and reliability of our approach, we emphasize its potential for real-time applications in urban monitoring.

We have revised the Introduction section to reflect these contributions and to better emphasize the points above. Therefore, we believe this revised section could provide a more comprehensive comparison with the existing literature and highlight the novel aspects of our work, situating it within the broader context of urban monitoring and critical event detection.

Editor’s Comment

b) Provide some suggestions on how the results can be used by stakeholders involved in emergency response planning.

Response

As already mentioned in the original version of the paper and further highlighted in the extended discussion section of the revised manuscript, the proposed approach presents multiple applications for urban stakeholders, even though it is not intended to function autonomously. Our solution is designed to serve as a complementary early warning system, augmenting other monitoring tools based on emergency call centers, surveillance cameras, and corroborative field reports traditionally employed by city stakeholders involved in urban security. Our methods can be seamlessly integrated into existing monitoring and emergency response planning systems. Consequently, stakeholders can leverage the large-scale, precise, and timely detection capabilities provided by NSD to enhance overall preparedness and response efficiency. In this context, it is worth mentioning that this work has been developed within a broader project (ANR DISCRET) aimed at developing a prototype warning system for crisis management, security, and emergency services during the Paris 2024 Olympic Games, which will involve multiple stakeholders for security and emergency response. Towards an improvement of the proposed solutions, we are currently investigating the possibility of enriching the output of the anomaly detection module presented in this paper with information extracted from X (former Twitter) social media. This integration aims to characterize and contextualize events more effectively, thus providing more comprehensive information and making better use of feedback from the population via channels not specifically dedicated to warning.

Concretely, potential applications of our contribution could therefore involve the following stakeholders:

 1. Emergency Response Services: The potential of NSD for rapid and precise identification of critical events can significantly improve the response time of emergency services by providing real-time alerts about potential critical events. This timely information allows for the rapid deployment of resources to affected areas, potentially saving lives and mitigating damage.

 2. Crisis Management Authorities: The integration of NSD and social media data can help crisis managers gain a more detailed understanding of unfolding events. This comprehensive situational awareness is crucial for making informed decisions and coordinating efforts among different emergency response units.

 3. Urban Security Agencies: For agencies responsible for maintaining public safety, the ability to detect anomalies at a finer spatial and temporal resolution enables more precise monitoring of high-risk areas. This capability can prove vital during large-scale events like the Olympic Games, where the potential for security threats is elevated.

 4. Local Government and Municipal Services: Local authorities can use the data to monitor urban dynamics continuously, allowing for proactive measures in traffic management, public transportation adjustments, and crowd control during critical events.

 5. Public Health Officials: In the event of public health emergencies, such as disease outbreaks or environmental hazards, the system can provide early warnings that facilitate rapid public health responses. The integration of various data sources ensures a comprehensive understanding of the situation, aiding in the containment and management of such crises.

The ability to contextualize and corroborate detected anomalies with social media, human judgment, and other data sources provides a robust foundation for proactive and informed decision-making by emergency planning stakeholders. This is now developed in the discussion section of the revised manuscript.

Editor’s Comment

 1. Please ensure that your manuscript meets PLOS ONE’s style requirements, including those for file naming. The PLOS ONE style templates can be found at:

 • PLOS One formatting sample main body

 • PLOS One formatting sample title authors affiliations

Response

We have taken into consideration all the journal requirements and made our paper compliant with them.

 • Style Requirements: We have ensured that our manuscript meets PLOS ONE’s style requirements, including the correct file naming conventions. We have utilized the PLOS ONE LaTeX template provided at the shared links for formatting the main body, title, authors, and affiliations. The manuscript has been revised to adhere to these guidelines to ensure consistency and compliance with PLOS ONE’s standards.

Editor’s Comment

 2. Please note that PLOS ONE has specific guidelines on code sharing for submissions in which author-generated code underpins the findings in the manuscript. In these cases, all author-generated code must be made available without restrictions upon publication of the work. Please review our guidelines at PLOS One materials and software sharing and ensure that your code is shared in a way that follows best practice and facilitates reproducibility and reuse.

Response

 • Code Sharing: We have reviewed the guidelines provided at the link for materials and software sharing and ensured that all author-generated code is shared in a manner that facilitates reproducibility and reuse. The code underpinning our findings has been uploaded to a publicly accessible repository on GitHub (https://github.com/licit-lab/discret). We have included the relevant links in the manuscript.

Editor’s Comment

 • “This work is supported by the French ANR research projects DISCRET (grant number ANR-19-FLJO-0002-01) and PROMENADE (grant number ANR-18-CE22-0008).”

 • “This work is supported by the French ANR research projects DISCRET (grant number ANR-19-FLJO-0002-01) and PROMENADE (grant number ANR-18-CE22-0008).

 • EG is the author awarded for the project DISCRET. AF is the author awarded for the project PROMENADE.

 • URL of the funder: https://anr.fr

Response

 • Funding Information: We have removed the funding information from the Acknowledgments section of our manuscript as requested. The Acknowledgments section has been revised accordingly. Furthermore, we have updated the Funding Statement to ensure it is correctly formatted and complete. The amended Funding Statement now reads:

 • “This work is supported by the French Agence Nationale de la Recherche projects DISCRET (grant number ANR-19-FLJO-0002-01) to AF, CZ, EG, NE, PL, SR, ZS, and PROMENADE (grant number ANR-18-CE22-0008) to AF. URL of the funder: https://anr.fr. The funders had no role in study design, data collection and analysis, decision to publish, or preparation of the manuscript.”

Editor’s Comment

 4. We note that you have indicated that there are restrictions to data sharing for this study. PLOS only allows data to be available upon request if there are legal or ethical restrictions on sharing data publicly. For more information on unacceptable data access restrictions, please see PLOS One data availability.

 • b) If there are no restrictions, please upload the minimal anonymized data set necessary to replicate your study findings to a stable, public repository and provide us with the relevant URLs, DOIs, or accession numbers. For a list of recommended repositories, please see PLOS One recommended repositories. You also have the option of uploading the data as Supporting Information files, but we would recommend depositing data directly to a data repository if possible.

Response

 • Data Sharing Restrictions: We note that the mobile phone data used in our study are proprietary and confidential to Orange. For this reason, we cannot share the full dataset publicly. However, for the sake of reproducibility, we have agreed in the Data Management plan to share a minimal subset of the data used in our analysis of the Notre Dame fire case study presented in the paper. This filtered and aggregated dataset has been made publicly available on the same GitHub repository used for code sharing, at the following URL: https://github.com/licit-lab/discret/tree/main/data-sample.

 • Concerning the Data Availability statement, we propose the following revision:

 • “Network signaling data are proprietary and confidential. We obtained access to these data from Orange France within the framework of the research project ANR DISCRET (ANR-19-FLJO-0002-01). For the sake of reproducibility of the research, a minimal subset is available at https://github.com/licit-lab/discret/tree/main/data-sample, as agreed in the project Data Management Plan. Access to the full dataset can be requested from Orange on a contractual basis, by contacting Dr. Thierry Nagellen at Orange Innovation/Research (Director of the Research Augmented Customers and Collaborators Domain) via email: thierry.nagellen@orange.com. Please note that Dr. Thierry Nagellen is not affiliated with this research as an author."

Editor’s Comment

 5

---

## [Decision Letter · Decision Letter 1]

6 Aug 2024

Early Detection of Critical Urban Events using Mobile Phone Network Data

PONE-D-24-11293R1

Dear Dr. Furno,

We’re pleased to inform you that your manuscript has been judged scientifically suitable for publication and will be formally accepted for publication once it meets all outstanding technical requirements.

Kind regards,

Luan Carlos de Sena Monteiro Ozelim, D.Sc.

Academic Editor

PLOS ONE

Additional Editor Comments (optional):

Reviewers' comments:

Reviewer's Responses to Questions

**Comments to the Author**

1. If the authors have adequately addressed your comments raised in a previous round of review and you feel that this manuscript is now acceptable for publication, you may indicate that here to bypass the “Comments to the Author” section, enter your conflict of interest statement in the “Confidential to Editor” section, and submit your "Accept" recommendation.

Reviewer #1: All comments have been addressed

Reviewer #2: All comments have been addressed

2. Is the manuscript technically sound, and do the data support the conclusions?

Reviewer #1: Yes

Reviewer #2: Yes

3. Has the statistical analysis been performed appropriately and rigorously? 

Reviewer #1: Yes

Reviewer #2: Yes

4. Have the authors made all data underlying the findings in their manuscript fully available?

Reviewer #1: Yes

Reviewer #2: Yes

5. Is the manuscript presented in an intelligible fashion and written in standard English?

Reviewer #1: Yes

Reviewer #2: Yes

6. Review Comments to the Author

Reviewer #1: The authors have meticulously addressed all the comments and suggestions provided, ensuring thorough revisions and comprehensive responses to enhance the overall quality and clarity of the manuscript.

Reviewer #2: the topic is quite interesting and the research is well reported. Moreover, All Question have been answered

7. PLOS authors have the option to publish the peer review history of their article (what does this mean?). If published, this will include your full peer review and any attached files.

Reviewer #1: No

Reviewer #2: **Yes: **Davide Micheli

---

## [Editor Report · Acceptance letter]

12 Aug 2024

PONE-D-24-11293R1 

PLOS ONE

Dear Dr. Furno, 

I'm pleased to inform you that your manuscript has been deemed suitable for publication in PLOS ONE. Congratulations! Your manuscript is now being handed over to our production team.

Kind regards, 

on behalf of

Dr. Luan Carlos de Sena Monteiro Ozelim 

Academic Editor

PLOS ONE